# Climate Change and Zoonoses: A Review of Concepts, Definitions, and Bibliometrics

**DOI:** 10.3390/ijerph19020893

**Published:** 2022-01-14

**Authors:** Walter Leal Filho, Linda Ternova, Sanika Arun Parasnis, Marina Kovaleva, Gustavo J. Nagy

**Affiliations:** 1European School of Sustainability Science and Research, Hamburg University of Applied Sciences, Ulmenliet 20, D-21033 Hamburg, Germany; w.leal@mmu.ac.uk (W.L.F.); marina.kovaleva@haw-hamburg.de (M.K.); 2Department of Natural Sciences, Manchester Metropolitan University, Manchester M15 6BH, UK; 3Birla Institute of Technology and Science, Pilani (BITS Pilani), Hyderabad Campus, Jawahar Nagar, Kapra Mandal, Medchal District, Hyderabad 500078, India; f20181113@hyderabad.bits-pilani.ac.in; 4Instituto de Ecología y Ciencias Ambientales (IECA), Facultad de Ciencias, Universidad de la República, Montevideo 11400, Uruguay; gnagy@fcien.edu.uy

**Keywords:** zoonosis, climate change, health hazards, humans, animals, bibliometric analysis

## Abstract

Climate change can have a complex impact that also influences human and animal health. For example, climate change alters the conditions for pathogens and vectors of zoonotic diseases. Signs of this are the increasing spread of the West Nile and Usutu viruses and the establishment of new vector species, such as specific mosquito and tick species, in Europe and other parts of the world. With these changes come new challenges for maintaining human and animal health. This paper reports on an analysis of the literature focused on a bibliometric analysis of the Scopus database and VOSviewer software for creating visualization maps which identifies the zoonotic health risks for humans and animals caused by climate change. The sources retained for the analysis totaled 428 and different thresholds (N) were established for each item varying from N 5 to 10. The main findings are as follows: First, published documents increased in 2009–2015 peaking in 2020. Second, the primary sources have changed since 2018, partly attributable to the increase in human health concerns due to human-to-human transmission. Third, the USA, the UK, Canada, Australia, Italy, and Germany perform most zoonosis research. For instance, sixty documents and only 17 countries analyzed for co-authorship analysis met the threshold led by the USA; the top four author keywords were “climate change”, “zoonosis”, “epidemiology”, and “one health;” the USA, the UK, Germany, and Spain led the link strength (inter-collaboration); the author keywords showed that 37 out of the 1023 keywords met the threshold, and the authors’ keyword’s largest node of the bibliometric map contains the following: infectious diseases, emerging diseases, disease ecology, one health, surveillance, transmission, and wildlife. Finally, zoonotic diseases, which were documented in the literature in the past, have evolved, especially during the years 2010–2015, as evidenced by the sharp augmentation of publications addressing ad-hoc events and peaking in 2020 with the COVID-19 outbreak.

## 1. Introduction

“Between animal and human medicine, there is no dividing line, nor should there be”.Rudolf Virchow (1821–1902)

Wildlife, humans and their domesticated pets, and the environment are closely connected via their various roles in maintaining and spreading contagious maladies. For example, for a long time, probably unreasonably, wildlife has been “accused” of being the source of zoonotic diseases among humans [1]. In any case, closer contact between humans and animals may lead to the spread of more viral or bacterial infections [2].

Zoonotic diseases were defined in 1951 by the Expert Committee on Zoonoses as “diseases and infections that are naturally transmitted between vertebrate animals and man” [3]. Nonetheless, the German physician and pathologist Rudolf Virchow introduced the term “zoonoses” at the end of the 19th century to describe human diseases shared with animals [4,5]. “Zoonoses” derives from ancient Greek (*zoon*: animals, and *noson*: disease). Besides, this term is considered as the most appropriate in comparison to “anthropozoonosis” (from animals to humans) and “zooanthroponosis” (from humans to animals), which are focused on the predominant path of transmission between humans and other vertebrates [4]. Nevertheless, such terms have been synonymously used for all diseases found in animals and humans. Additionally, a third word, “amphixenoses”, has been coined to describe infections that can be transmitted in either direction and maintained in humans and lower vertebrate animals [6].

A large variety of pathogens have been transmitted from animals to humans. The contact between an animal and a human needs to occur first before a pathogen can become a threat to a population [7]. A pathogen is “an organism causing the disease to its host, with the severity of the disease symptoms referred to as virulence. Pathogens are taxonomically widely diverse and comprise viruses and bacteria as well as unicellular and multicellular eukaryotes”. [8]. Among the types of zoonotic pathogens mention may be made to *Salmonella* spp., *Campylobacter* spp., and *E. coli* pathotypes, among others. In many cases, infection takes place by the fecal-oral route (e.g., by food and water) and by direct animal contact. 

An emerging infectious disease (EID) is an infectious disease that “either has appeared and affected a population for the first time, or has existed previously but is rapidly spreading, either in terms of the number of people getting infected or to new geographical areas” [9].

There are various other factors associated with the emergence of zoonoses, such as globalization, international trade, land-use changes, and, increasingly, climate change associated with vector-borne zoonoses. Changing climate conditions may also be associated with the spread of hantaviruses through impacts on the hantavirus reservoir host populations.

To date, over 200 types of zoonotic diseases are recognized, which account for a substantial proportion of novel and current human illnesses [10]. Furthermore, it is acknowledged that around 60% of all human pathogens and 75% of emerging infectious diseases originate from animals [11]. 

Evidence suggests that up to 60,000 annual deaths are attributable to Rabies, and other zoonoses, i.e., Avian flu, Ebola virus, or Rift Valley fever, pose additional risk. It is worth stating that such illnesses have an impact on both human and animal health, leading to reduced yield (e.g., quality and safety of milk or eggs) or death, and thereby affecting the livelihoods of farmers and the economies of nations [12]. At the global level, the 13 most widespread zoonoses had a significant impact on poor livestock workers in low- and middle-income countries by causing 2.4 billion disease cases and 2.7 million deaths per annum in humans [13]. According to experts, zoonoses could jeopardize human health in various ways: *Endemic zoonoses* are widespread among poor populaces and cause billions of illnesses and millions of fatalities each year. Examples of endemic zoonoses include cysticercosis, brucellosis, bovine tuberculosis, leptospirosis, and foodborne zoonoses.*The outbreak of epidemic zoonoses* usually occur infrequently, and anthrax, rabies, Rift Valley fever, and leishmaniasis are only a few examples. However, they may emerge in vulnerable populations due to climate change, flooding, reduced immunity, famine, or illness. Outbreaks show a high degree of temporal and spatial changeability [14].*Emerging zoonoses* might again emerge in a population or have occurred earlier, but currently, they are quickly increasing in terms of incidence or geographical area [14,15]. Two-third of all new and emerging infectious diseases is zoonoses [16]. Based on the evidence between 1940 and 2004, there have been reported around 335 cases of such events [17]. Moreover, it is suggested that COVID-19, caused by SARS-CoV-2, is to be classified under emerging infectious diseases (EIDs) of possible animal origin since no animal reservoir has yet been identified [18].*Zoonoses* were initially zoonotic, but nowadays, they are mainly or exclusively spread via human-to-human transmission. Diseases such as HIV/AIDS, pneumonia, malaria, measles, and dengue fever are included. Their current intensity is roughly comparable to endemic zoonoses (practically all because of HIV/AIDs) [14].

### 1.1. Classification of Zoonotic Diseases

Zoonoses are categorized according to their etiological agents (i.e., bacterial, viral, parasitic, mycotic/fungal zoonosis), reservoir hosts (whether a human or an animal), or the life cycle of pathogens (epidemiological classification) [4,6,13]. Figure 1 presents some of the main groups of zoonoses based on their etiological agents. 

On the other hand, the afore-mentioned terms of “anthropozoonosis”, “zooanthroponosis”, and “amphixenoses” are used within the reservoir host context by also defining the direction of infection [6] (Figure 2).

Accordingly, in the light of epidemiological classification based on the zoonosis maintenance cycle, zoonotic diseases are divided into four groups: direct zoonoses (orthozoonoses), cyclozoonoses, pherozoonose (metazoonoses), and saprozoonoses [4].

*Direct zoonoses (orthozoonoses)* are spread from an infected vertebrate host to a vulnerable vertebrate host via direct contact, contact with a fomite, or mechanical transmission. During such a process, the agent experiences little or no propagative modification and no significant developing alterations. Rabies, trichinosis, and brucellosis are only a few examples.*Cyclozoonoses* require a couple of vertebrate host species but no invertebrate host to complete the agent’s evolution cycle. Typical cyclozoonoses are *Human taeniasis*, Echinococcosis and Pentastomid infections.*Invertebrate vectors transmit pherozoonose (metazoonoses)*: the agent multiplies, evolves, or both in the invertebrate, and there is always an inherent incubation period (prepatent) before transmission to another vertebrate host, wherever possible. Arbovirus infection, plague, and schistosomiasis are only some of many examples.*Saprozoonoses* have a vertebrate host and a non-animal developmental site or reservoir, i.e., food, soil, and plants. Examples include the various forms of larva migrants and some of the mycosis [6].

As defined by the WHO, Zoonosis refers to “diseases and infections that are naturally transmitted between vertebrate animals and man”, or in simple terms, an infectious disease with the potential to transmit from non-human animals to humans. Zoonoses, which have links with some phenomena such as climate change [19], has particular relevance in current studies due to recent COVID-19 and Ebola outbreaks [10,20]. According to the CDC, almost six out of every ten infectious diseases can be spread from animals; three out of every four emerging infectious diseases originate from animals. Their spread is facilitated by the environment, water, food, direct and indirect contact, or vectors [21]. Arthropods are the most significant transmission vectors due to their high adaptability, plasticity, and abundance [22]. Furthermore, the acceleration of zoonotic pathogens is attributable to changes in climate and ecology due to human impact and vector transportation rates that are faster than their incubation periods [23].

Furthermore, societies residing in and around tropical regions are disproportionately affected by zoonotic diseases. Effective management and sound ecological understanding of zoonotic disease systems can prevent many infections from spreading to human beings, especially conditions involving vectors such as ticks [24], which also notes that management may be increasingly compelling when targeted at the root cause. Interventions targeting vectors and pathogen transmission via wildlife hosts may be more effective than traditional intervention strategies such as vaccination, especially for neglected tropical diseases where the ecological complexity of disease spillover is ignored. Diseases such as Rabies, leprosy, and Leishmaniasis primarily affect disadvantaged populations in low-income settings. The effect of zoonoses in developed and developing countries is significantly different. For example, Chagas disease, easily controllable via pharmaceutical and vector preventive measures, still has high incidence rates in Central and South America [22]. 

The avian influenza of 2005 was instrumental in uniting global bodies to address the zoonotic threat. One health as a concept approaches human, ecosystem, and animal health as an interconnected network rather than addressing each sector individually, fostering a robust and collaborative environment to manage health assessments and prevent zoonotic diseases while considering human medical and veterinary practices [25]. To facilitate the study of zoonotic pathogens and ensure adequate public health response to zoonotic transmission, the WHO/FAO classified zoonoses in 1967 according to reservoir hosts harboring them [21]:(A)Anthropozoonoses—Hosts are lower animals;(B)Zooanthroponoses—Hosts are human beings;(C)Amphixenoses—Hosts are lower animals as well as humans.

However, other classification systems developed later were more valuable and instructive. Hence, primarily, three methods of the classification of zoonoses are widely studied and used. 

(1)Environment

Zoonoses can be classified according to the environment where the infectious agents circulate.

*Synanthropic:* The infectious agents’ cycle is restricted to domestic animals, for example, bovine tuberculosis, ringworm, and listeriosis. The transmission of these diseases occurs aerogenically through conjunctiva or percutaneously;*Exoanthropic:* The infectious agents’ cycle is within feral or animals living outside domestic boundaries, for example, tick-borne encephalitis or arboviruses. The transmission of infection from an animal to a human usually takes place through a hematophagous vector [26].

(2)Etiology

According to etiology, etiological agents of zoonoses are classified as bacterial (tuberculosis, Lyme disease), viral (AIDS, Ebola), fungal (ringworm), parasitic (giardiasis, malaria), and metazoan (anisakidosis). In addition, the agent’s description considers factors such as its pathogenicity, invasivity, and toxigenicity. Furthermore, to be classified as the etiological agent, Koch’s postulates must be fulfilled [27]:(A)Must be detected in all cases of the diseases;(B)Must be isolated and cultured in the laboratory;(C)Must be able to produce the disease in another host after laboratory cultivation;(D)Recovered from the same inoculated host;(E)Production of antibodies in the inoculated host.
(3)Zoonotic Cycle

Zoonoses are also classified according to their maintenance cycle. This classification is significant in the treatment and control of zoonoses [4,28].

*Cyclozoonoses:* The infectious agent life cycle requires more than one non-human vertebrate to be completed. Here, the non-human vertebrate acts as the intermediate host and harbors the infectious agent, which ultimately infects humans. Some examples include human taeniasis and Hydatid cyst disease. These zoonoses are relatively rare.*Saprozoonoses:* The life cycle of the infectious agent requires non-living matter, such as organic matter in the soil or water and non-human vertebrate hosts. However, direct infections, in this case, are rare. The sapronotic agents can multiply using two different life cycles either through a saprophytic life cycle via an abiotic agent or parasitically through a vertebrate host. Examples include swine erysipelas and actinomycosis.*Orthozoonoses:* They are also known as direct zoonoses. The vector transmission occurs through an infected to a susceptible vertebrate directly or via a rabies mechanical vector.*Pherozoonoses:* The vectors require both vertebrates and invertebrates for the completion of the life cycle. The vector multiples or develops in the invertebrate to spread, for example, arbovirus, plague, Lyme disease, and encephalitis.

### 1.2. Climate Change and Zoonoses

Zoonoses refer to the transmission of diseases/infections from animals to humans. Statistics indicate that six out of ten infectious diseases arise from animal to human transmission with global prevalence [25]. Such transmissions may occur in urban or rural areas [29].

Zoonotic diseases, such as those spread by mosquitoes and other related vectors, have increased in recent years. The rise in global temperatures has created favorable conditions for breeding specific vectors, especially in poorly developed countries [30]. Additionally, the increased precipitation and parallel flooding conditions in certain areas provide the perfect breeding grounds for vectors. Flooding conditions amplify the possibility of waterborne disease transmission [31]. 

In Russia, climate change significantly affects the spread of diseases. The increased incidence of zoonotic disease is attributed to fluctuating temperatures, causing prolonged periods of vegetation growth and broadening habitat availability, thereby allowing zoonotic pathogens and related vectors to have more suitable living conditions that promote survival and reproduction. Furthermore, increased temperature is causing the melting of permafrost in the area, leading to the concern that the permafrost degradation will expose ancient human burial sites and result in the revival of vectors that spread deadly infections [32].

Climate change can also influence the geographical distribution of insect spread diseases [33]. Zoonotic viruses previously localized to areas with high temperatures, such as the tropics, have been observed worldwide, having spread to subtropical climates and areas with high altitudes. Climate change has caused temperature changes in various geographical regions, meaning that areas previously free from certain diseases now see rises in infection prevalence. Furthermore, climate change is causing people’s general health to deteriorate, making it easier for zoonotic infection to spread [33], as seen with the Zika and dengue viruses, which are now global threats [34,35]. Additionally, in less developed areas, the increase in droughts and flooding has cut freshwater availability, which causes humans to ingest water contaminated with zoonotic waterborne diseases, such as schistosomiasis [36,37,38].

Aside from this, the changes in climatic conditions have forced pathogens and vectors to develop adaptation mechanisms. Such development has resulted in the diseases becoming resistant to conventional treatments due to their augmented resilience and survival techniques, thus further favoring the spread of infection [32]. In some contexts, changes in climate conditions may help to increase the resistance of microbes, such as bacteria and viruses, making treatment more difficult and contributing to an increase in the disease spread.

## 2. Materials and Methods

We performed a bibliometric analysis by focusing on the quantitative appraisal of the published literature regarding zoonotic health risks on human and animal health due to climate change effects. The bibliometric concept was introduced in the late 60 s by Alan Pritchard, who defined bibliometrics as “the application of mathematical and statistical methods to books and other media of communication” [39,40]. In addition, while performing such an analysis method, we considered the use of the Scopus database as more appropriate since its main advantage is to provide the most comprehensive coverage of scientific, technological, medical, and social disciplines currently available [41]. 

Considering the searching method at Scopus and trying to extend our search query, we used: zoono* AND “climat* chang*” OR “climat* variab*” AND health, within TITLE-ABS-KEY. Next, we narrowed down our search by limiting to documents in their final stage of publication; from 2000 to 2020 (the year 2021 was excluded as it is not completed yet); in the English language; whose source type were journals, books, or book chapters; and including the following subject areas: Medicine; Immunology and Microbiology; Agricultural and Biological Sciences; Veterinary, Environmental Science; Biochemistry, Genetics and Molecular Biology; Multidisciplinary; Social Science; Pharmacology, Toxicology and Pharmaceutics; and Earth and Planetary Sciences. This way, we excluded the subject areas not relevant to our topic due to possible false-positive results. 

The information obtained from the search query was exported to Microsoft excel and double-checked to eliminate the duplicates. Lastly, the selected data from Scopus (as of 13 July 2021) were exported to the VOSviewer database to generate network visualization maps. Because bibliometric analysis is a rigorous method, we followed precise steps, as defined in previous papers on the bibliometric approach [42]. Therefore, by summarizing all the steps taken during our research, we obtained the following overview, illustrated in Figure 3.

Literature reviews are considered one of the best ways to gather knowledge regarding a particular field and further analyze it to come to conclusions [43]. The bibliometric analysis serves as a statistical tool to extract information from a defined set of literature [44]. Traditionally, they consisted of scientific productions or specific, highly cited publications subdivided as lists of prominent authors with publications or national or subject bibliographies [45]. Further extraction and manipulation of data are carried out using tools such as citation or content analysis [46]. Other tools include co-citation analysis, keyword analysis, and bibliographic linking quotation, which can be used according to the researcher’s needs [47]. Bibliometric analysis is distinguished from a review paper in the sense that a review paper determines the progress of a field and elaborates on its challenges, whereas a bibliometric analysis determines the trends of a field based on the literature database selected [48].

### Search Strategy and Data Export and Analysis

The data was obtained using the Scopus database that has been extensively used for bibliometric analysis, considering that it is one of the largest databases available. Scopus is also inclusive of PubMed [49]. It also indexes multiple components of a scientific publication, such as publication title, abstract, and author name. Scopus has been used for numerous large-scale evaluations, enhancing its legitimacy as a bibliometric analysis database [50]. 

The search string used was TITLE-ABS-KEY (“climate change” AND “zoonosis”) AND (LIMIT-TO (SRCTYPE, “j”)) AND (LIMIT-TO (DOCTYPE, “ar”) OR LIMIT-TO (DOCTYPE, “re”)). The search category limited the data to be sourced from journals and is of article or review article type to ensure sound analysis. A total of 339 articles fit the search category from the period 2000 to 2021. The data were exported in CSV format to Microsoft Excel (Microsoft, Redmond, WA, USA). Further analysis and formation of network visualization maps were also carried out using VOSViewer software. 

## 3. Results

Initially, by only applying the keywords within TITLE-ABS-KEY, the search query identified a total of 535 publications (TP, total publications). However, after limiting our search by applying the inclusion above and the exclusion criteria, the retrieved data consisted of 432 documents. In addition, four publications appeared twice in the excel list; therefore, we made the necessary adjustments by removing the double documents from the list to have an accurate result. 

### 3.1. Documents by the Publication Year

The number of publications per annum appeared to be not significant from 2000–2008. Afterwards, from 2009–2015, a fluctuated growth of publications can be observed. However, after 2015 and onwards, the trend increased significantly, explicitly in 2020, where the number of publications accounted for the highest number (*n* = 76). Successively, Figure 4 depicts the growth tendency of publications about zoonotic health risks-related literature due to climate change. 

The peak of publications coincided with humanity’s situation with the COVID-19 pandemic, whose first cases appeared in December 2019 in Wuhan (China). COVID-19 was considered a zoonotic disease, and bats were believed to be the reservoir hosts for SARS-CoV-2 [51]. Still, the upward trend seems to be in line with the growing public health threat of zoonotic diseases, highlighted in the guide launched on 11 March 2019, by the Food and Agriculture Organization of the UN (FAO), the World Organization for Animal Health (OIE), and the World Health Organization (WHO), in respect to countries that take a One Health approach to tackle zoonotic disease risks [52]. Notably, with the beginning of the 21st century, the WHO’s Eastern Mediterranean Region would be recognized as a hot spot concerning emerging zoonoses [53]. 

### 3.2. Document by Source

Regarding the most preferred Journals for publication, Veterinary Parasitology has the most publications (*n* = 15), followed by Parasites and Vectors (*n* = 13) and Vector-Borne and Zoonotic diseases (*n* = 11), OIE Revue Scientifique et Technique, and Plos Neglected Tropical Diseases (*n* = 9), and International Journal for Parasitology (*n* = 7). Some journals, i.e., Veterinary Parasitology and OIE Revue Scientifique, have a long history in publishing on such a topic compared to other journals. Their first publication on this matter dates back to 2004, in contrast to the Parasites and Vectors journal, whose publications started a decade later.

Most of the documents of these top five sources were published during 2013–2019, peaking in 2013 and 2017. Interestingly, most of the articles published in 2010 focused on the control and epidemiology of leishmaniasis and leptospirosis and the effect of climate and ecological changes on vector-borne diseases, such as hantavirus and Crimean-Congo hemorrhagic fever. Most of the articles published in 2013 focused on the interactions of the environment and ecology with zoonotic parasites and One Health approaches to public health. However, the percentage of publications from these top five sources decreased from 2018–2020 to 7%, partly in line with the sharp increase of documents. 

The trends from 2017–2019 show that regional studies examining the transmission of zoonotic diseases and environmental causes have steadily increased.

### 3.3. Documents by Affiliations and Country

The list of top ten affiliation institutions actively participating in publishing documents in the topic was led by the Centers for Disease Control and Prevention (*n* = 16), followed by the University of Queensland and London School of Hygiene and Tropical Medicine (*n* = 10, respectively), and Harvard University, Københavns Universitet, and Friedrich-Loeffler-Institute (*n* = 9, accordingly) (Table 1).

Additionally, the United States ranked first with 130 publications (30%), preceding the United Kingdom with 71 documents (16.4%), followed by Australia and Canada with 40 (9.3%) and 38 (8.8%) publications, respectively (Table 2).

### 3.4. Documents by Type

Regarding the document type, journal articles represented the most significant proportion of literature written on zoonotic health hazards to humans and animals due to climate change by 44.2% (*n* = 191), to be followed by other categories, i.e., reviews by 35.6% (*n* = 154), book chapters by 6.5% (*n* = 28), and editorials by 4.5% (*n* = 21).

### 3.5. Citation Index

Out of the total number of documents examined for the h-index, 60 of them were minimally cited 60 times. On the other hand, the paper named “Present and future arboviral threats” [54] is the most cited document among 432 identified documents related to our topic of interest. Such document was published by the Antiviral Research Journal in 2010 and cited 831 times. The two other most cited articles are “Safeguarding human health in the Anthropocene epoch: report of the Rockefeller Foundation-Lancet Commission on planetary health” [55], published by The Lancet in 2015, and “Foodborne diseases. The challenges of 20 years ago persist while new ones continue to emerge” [56], published by the International Journal of Food Microbiology, cited 740 and 660 times, accordingly.

### 3.6. Co-Authorship Analysis

In terms of the co-authorship analysis, by using “Countries” as a unit of analysis, we set the criteria of a minimum number of documents of a country to be five. As a result, out of 132 countries, only 17 met the threshold. Therefore, the United States led the list, with the highest number of documents (*n* = 56) and citations (*n* = 5087). The second country on the list was the United Kingdom (31 documents and 3915 citations). Finally, Canada and Australia had the same number of documents (*n* = 18); however, Canada led with 1895 citations compared to Australia with 1314 citations (Table 3).

As visualized from the VOSviewer (Figure 5), four clusters were identified: (a) the green cluster consisting of the United States, United Kingdom, China, Australia, and Kenya; (b) the red cluster consisting of Italy, Germany, Spain, New Zealand, Switzerland, France, and The Netherlands; (c) the blue cluster consisting of Canada, Norway, and Denmark; and (d) the yellow cluster consisting of Sweden and Thailand. In this context, countries under the same cluster signify their vital interests in the same field of research. Referring to the large node, the United States and the United Kingdom had the highest proportion of published documents related to our topic. In addition, the distance between two nodes representing the USA and the UK is relatively small, which indicates an intense collaboration between those two countries. On the other hand, countries like Australia, China, and Kenya, though positioned under the green cluster, have fewer publications compared to the two leading countries.

Similarly, some collaboration is less intense or sometimes missing, as in Australia and Kenya. Furthermore, Italy and Germany (under the red cluster) are the leading countries in the European region for the number of documents (larger size node) with international researchers. By contrast, Italy showed intense research collaboration with The Netherlands, France, and Switzerland (smaller distance between nodes representing these countries). In contrast, Germany collaborated with other countries, i.e., Spain and The Netherlands, rather than other EU countries. 

### 3.7. Co-Occurrence Analysis

Regarding the co-occurrence analysis, we used two units of analysis: “author keywords” and “index keywords”. Therefore, when applying the “author keyword” unit and setting five as the minimum number of occurrences of a keyword, 14 met the threshold out of 581 keywords. After identifying a pair of terms with identical meaning and excluding that term with the smallest number of occurrences, we obtained a list of 13 author keywords. Therefore, terms such as “climate change”, “zoonosis”, and “epidemiology” appeared to be the most frequent “author keywords” used by researchers in publications related to zoonotic health threats for humans and animals triggered by climate change (Table 4).

The VOSviewer visualization (Figure 6) shows 3 clusters: red comprising six items (climate change, vector, climate, transmission, wildlife, mosquitoes), green comprising four items (zoonosis, one health, West Nile virus, bacteria), and blue comprising three items (epidemiology, public health, and ecology).

When applying the “index keyword” unit and setting ten as the lowest number of occurrences of a keyword, 85 met the threshold out of 2390 keywords. After detecting identical terms and excluding those terms with the smallest number of occurrences, we ended up with a list of 80 index keywords. Table 5 displays the top ten index keywords per each of the identified clusters. Overall, the terms “zoonosis”, “human”, “climate change”, “non-human”, and “animals” resulted as the most frequent index keywords in publications related to our topic.

The occurrence (nodes) of index keywords in the retrieved literature visualized in the VOSviewer map of author keywords (Figure 7) shows that “human”, “climate change”, “animals”, “non-human”, and “zoonosis” have the highest frequency versus other index keywords. Four main clusters were found: (a) red (in the upper left of the map), consisting of 29 items and whose most frequent index keywords were climate change, zoonosis, article, public health and disease transmission; (b) blue (upper right), consisting of 16 items and whose most frequent index keywords were animals, non-human, parasitology, transmission, isolation and purification, and microbiology virology; (c) green (centered-bottom), consisting of 19 items and whose most frequent index keywords were Africa, animal health, Asia, deforestation, disease vectors, domestic animal, epidemiology, and Europe; and (d) yellow (centered-left), consisting of 16 items and whose most frequent index keywords were human, review, risk factor, epidemic, infection control, wildlife, communicable diseases, emerging, and environmental factor.

### 3.8. Citation Analysis

In terms of citation analysis, by using “authors” as the unit of analysis, after setting the minimum number of three documents from one author, six out of 740 authors met the threshold. Instead, by applying “documents” as the unit of analysis, after setting ten as the minimum number of document citations, 124 met the threshold out of the 208 documents. The most extensive set of connections offered by VOSviewer was of 34 documents, as some of the 124 identified ones were not linked to each other. 

Concerning bibliographic coupling, that is to say, how close or far authors are bound to cite similar publications, we chose “authors” as the unit of analysis. After setting three as the minimum number of an author’s documents, it resulted that six of the 740 authors met the threshold. 

Scopus’s CiteScore is a metric used as an alternative to Impact Factor produced by the Web of Science. Impact Factor uses articles and reviews as citable documents from the preceding two years to calculate the impact factor. In contrast, we calculated CiteScore to use all possible citable documents (articles, reviews, conference papers, notes) from the preceding four years to calculate the impact factor. 

Although the CiteScore metric serves to compare sources, it has disadvantages of its own. For example, CiteScore is not indicative of the reach a journal has to its audience and how much a journal has contributed to the progress of a field.

According to the CiteScore of 2020, Epidemiology and Infection, International Journal for Parasitology, PLoS Neglected Tropical Diseases, Parasites and Vectors, and Viruses journals had a CiteScore of 5 and above, with PLoS Neglected Tropical Diseases having the highest score of 7.1 and OIE Revue Scientifique et Technique having the lowest of 1.3. These data align with the change in the most preferred journals over the last few years, as commented in Section 3.2, Documents by source. 

### 3.9. Leading Institutions and Countries

The leading institutions are mainly located in North America, Europe, or Australia (e.g., The Centre for Disease Control and Prevention, USA, Friedrich-Loeffler-Institute, Germany, Agence de la santé Publique du Canada), indicating that the research output from Asia and Africa has been comparatively lower in this sphere. Moreover, the sector itself is interdisciplinary and lies within the sphere of the importance of public health. However, only the T.H. Chan School of Public Health is seen in the top 10 universities. 

The highest number of documents, 81, is produced by the United States of America, followed by the United Kingdom at 51 and Canada at 34 documents, which are the critical research countries. China is the only country from Asia among the top ten countries, indicating the potential for increasing research output and representation from Asia, Africa, and South America. 

The network visualization mode is used in the VOSViewer software to create a bibliometric map of the association of countries based on co-authorship. The minimum and the maximum number of documents per country are set to five and 25. Of the 102 countries available, 29 fit into these criteria, creating a bibliometric map (Figure 8) where four distinct clusters are observed. The red cluster includes Germany, Spain, France, and Romania; the blue one includes the United States, the UK, China, and Kenya; the yellow cluster includes Australia, Mexico, and Brazil; and the green cluster includes Denmark, Sweden, and South Africa. In each cluster, the distance between every country and the thickness of the association line determines the magnitude of collaboration and relatedness between the countries. The total link strength is highest for the USA at 74, followed by the UK at 68, Germany at 52, and Spain at 49. The total number of citations is highest for the United States, 4006, followed by the United Kingdom at 3060 and Germany at 1168. 

As of 16 June 2021, the leading authors (by publications and citations from the Scopus database) are from the USA, Spain, the UK, Italy, Canada, Kenya, and Belgium. Notably, three of the ten leading authors are affiliated with the same institution, the University of Valencia, Spain.

Regarding author keywords, the criteria set for the minimum number of times a keyword occurred was five; hence, out of the 1023 keywords, 41 met the requirements. After removing repeated words and synonyms, a total of 37 keywords were used to form the bibliometric map with nine clusters and 185 links in total (Figure 9). For example, the first cluster consists of seven items (infectious diseases, emerging diseases, disease ecology, one health, surveillance, transmission, and wildlife); the second cluster consists of six items (climate change, Africa, disease, globalization, parasites, and public health); the third cluster six items (zoonosis, West Nile virus, vector-borne diseases, *Leishmania infantum*, epidemiology, and *Dirofilaria repens*); and the fourth cluster seven items (ecology, Europe, risk assessment, tick, vector-borne, vectors, and Rift Valley fever). The other five clusters include three (arctic, COVID-19, and zoonotic diseases); two (leptospira and leptospirosis); two (climate and zoonotic cutaneous leishmaniasis); two (arbovirus and emerging infectious diseases); and one item (health). “Zoonosis” has the highest occurrence (68) followed by “Climate change” (66).

## 4. Discussion

Considering that

Zoonoses transmission of infections from animals to humans reaches 60% of the total infectious diseases [25].Zoonoses nowadays are mainly or exclusively spread via a human-to-human transmission (e.g., HIV/AIDS and dengue fever); their current intensity is roughly comparable to endemic zoonoses (practically all because of HIV/AIDs) [13].Climate change causes variations in temperature, rainfall patterns, and climate disaster incidence, influencing the global population’s transmission of diseases and affecting the transmission rates of various vector-borne diseases [30,31].Global warming favors the increase of vector-borne diseases, especially in poorly developed countries [30].The rainfall and flooding increase, in many world regions’ favor, the possibility of waterborne diseases transmission [30].Climatic changes forced pathogens and vectors to develop adaptation mechanisms facilitating the spread of infections [32].The avian influenza of 2005 highlighted the interconnected animal, environmental, and human health [25].The northern regions were not much affected by zoonotic diseases in the past; however, recent evidence suggests that climate change has accelerated the spread of infections in northern countries. For instance, climate change has resulted in the early onset of winters characterized by increased wet conditions, correlated to the increased spread of diseases via the amplified intensity of pathogens rather than increased host abundance or distribution. This indicates that the global climate crisis also can alter the nature of zoonotic infections through its effect on temperature and precipitation [57].

Zoonoses are a global threat and, like many infectious diseases in the category, can reach pandemic status, including H1N1 influenza and the more recent pandemic of COVID-19 [25,35].

The evolution of the published literature on zoonosis is shown in Figure 4 as well as the inflexion observed since 2015 and the changes in the percentage of publication or citations of the top sources from 2018–2020 seem to follow the facts mentioned above, with a net inflexion in 2020 in parallel with the COVID-19 pandemic.

## 5. Conclusions

Zoonotic diseases are an increasing global environmental problem. To date, zoonotic diseases, mainly the emerging ones, account for a substantial proportion of novel and current human illnesses. 

Zoonoses are a global threat because they can become a pandemic, as seen in the case of COVID-19. In addition, the global climate crisis influences the transmission of diseases among the world’s population, including the transmission rates of several vector-borne diseases, mainly in the tropical low- and middle-income countries. In recent years, changes in the timing and magnitude of temperature and rainfall have also accelerated the spread of infectious diseases in northern countries.

The quantitative bibliometric analysis of the Scopus database focused on zoonoses and climate, climate change or climate and variability, and health from 2000 to 2020 and was exported to the Vosviewer database to generate network visualization maps.

From the total analyzed publications (*n* = 428), the critical quantitative findings are as follows:The number of published documents increased in 2009–2015, peaking in 2020 with almost 18% of the total coinciding with the COVID-19 outbreak.The primary sources are periodicals, such as the Parasites and Vectors journal, Veterinary Parasitology, OIE Revue Scientifique et Technique, and Plos Neglected Tropical Diseases. Most of the documents were published from 2013–2019, peaking in 2013 and 2017 and sharply decreasing onward; however, PLoS Neglected Tropical Diseases reached the highest CiteScore in 2020.The top affiliation of authors is the Centers for Diseases Control and Prevention (USA). The USA, the UK, Australia, Canada, Germany, Italy, and Spain account for 85% of the authors.Journal articles were the primary document type (44.2%), followed by reviews (35.6%) and book chapters (6.5%).Sixty documents met the 60 citations (h-index) threshold; the most cited article was “Present and future arboviral threats” [54], published in the Antiviral Research Journal (*n* = 831).From the 132 countries included in the co-authorship analysis, 17 met the threshold (*n* = 5), led by the USA and followed by the UK, Canada, Australia, Italy, Germany, France, Sweden, Spain, and The Netherlands. Additionally, the visualization with the Vosviewer map showed four clusters sharing similar research interests. The largest node includes the USA, the UK, China, Australia, and Kenya. Italy and Germany led in the second-largest cluster, mainly from Europe plus New Zealand.The co-occurrence analysis “author keywords” showed that 13 keywords (out of 581) exceeded five occurrences; the top four were “climate change”, “zoonosis”, “epidemiology”, and “one health”. The VOSviewer map showed three clusters, the most prominent one including climate change, vector, climate, transmission, wildlife, and mosquitoes. Eighty keywords met the “index keyword” threshold (*n* = 10), with human, climate change, animals, non-human, and zoonosis being at the top.The citation analysis by authors (threshold: *n* = 3) and documents (*n* = 10) showed that only six authors (out of 740) and 58% of documents (out of 124) met the threshold.The USA, the UK, Germany, and Spain led the link strength (inter-collaboration), while the leading authors are from the USA, Spain, the UK, Italy, and Canada.The author keywords showed that 37 out of the 1023 keywords met the threshold (*n* = 5). The bibliometric map shows nine clusters and 185 links. The largest one contains seven items (infectious diseases, emerging diseases, disease ecology, one health, surveillance, transmission, and wildlife).

This paper has some limitations. One of them is the fact that literature was analyzed over a limited period of time. Secondly, only literature in English was considered. Moreover, the paper did not consider case studies to illustrate specific findings. These may be the subject of further studies.

However, despite these limitations, the paper provides a welcome addition to the literature since zoonoses may increase in incidence as a result of changing climate conditions, and greater interference with natural ecosystems, putting humans in more direct contact with animals.

In conclusion, there are three critical findings deriving from this review paper. Firstly, zoonotic diseases have become a global crisis beginning around 2010–2015, evidenced by the sharp augmentation of publications peaking in 2020. This is associated with the COVID-19 outbreak, which has drawn renewed attention to the problem. Secondly, all the bibliometric indicators show that the USA, the UK, Canada, Australia, Italy, and Germany perform most of the published research. Thirdly, the predominance of animal health-focused journals as outlets for articles on zoonoses has changed. Many other journals are now focusing on zoonoses, which may be explained by the increased interest in animal to human diseases transmission.

## Figures and Tables

**Figure 1 ijerph-19-00893-f001:**
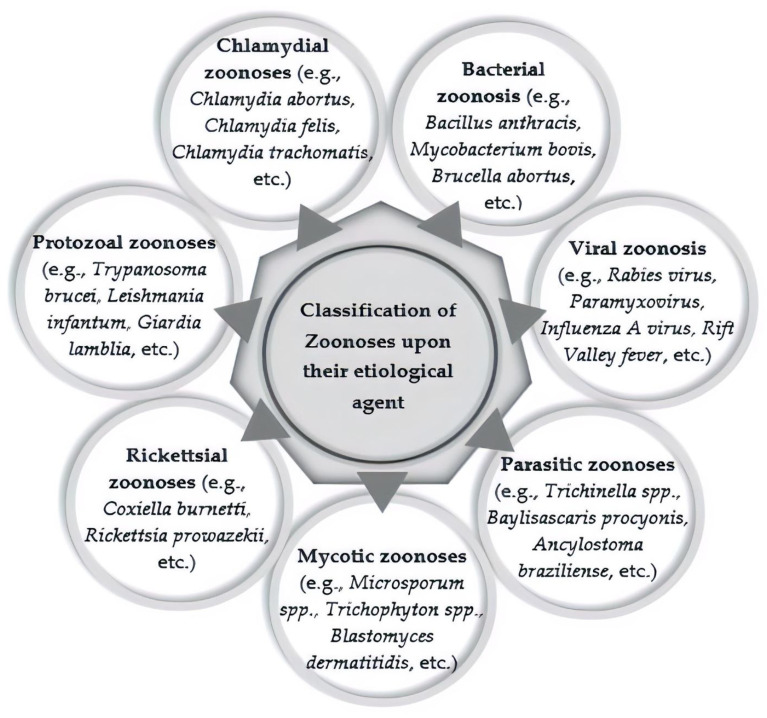
Classification of zoonoses according to their etiological agents (main groups).

**Figure 2 ijerph-19-00893-f002:**
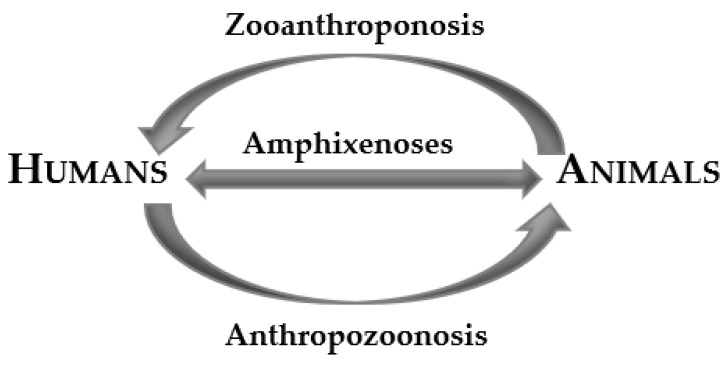
Classification of zoonoses according to the direction of infection.

**Figure 3 ijerph-19-00893-f003:**
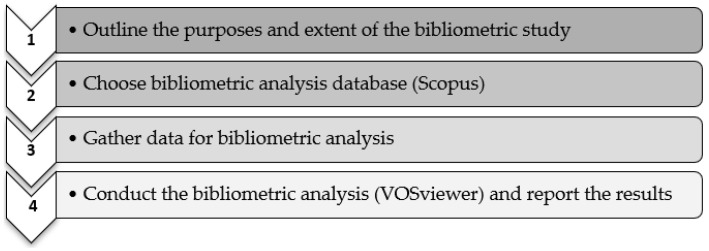
Steps followed in our research study.

**Figure 4 ijerph-19-00893-f004:**
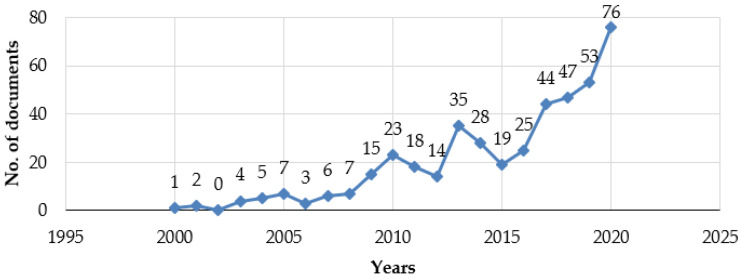
Documents by year.

**Figure 5 ijerph-19-00893-f005:**
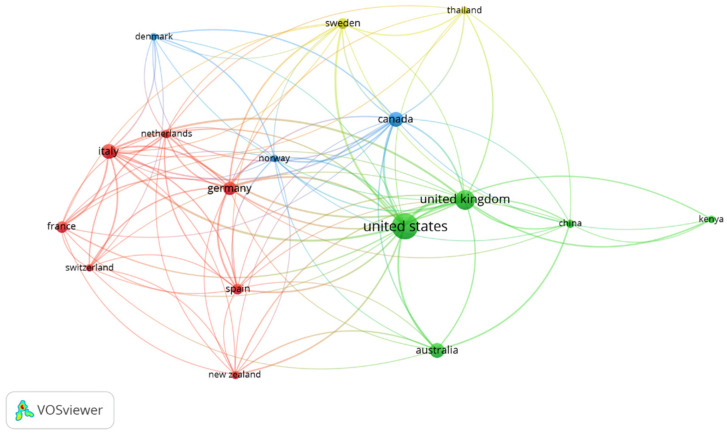
VOSviewer visualization of co-authorship by countries.

**Figure 6 ijerph-19-00893-f006:**
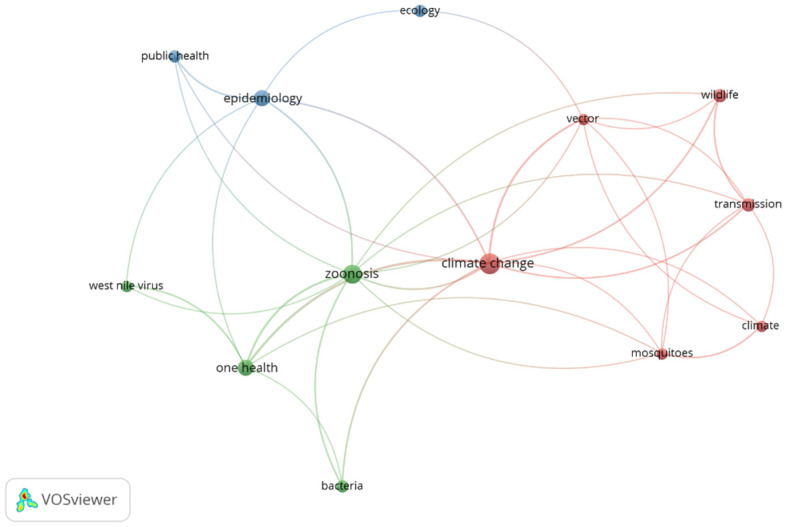
VOSviewer visualization of author keywords.

**Figure 7 ijerph-19-00893-f007:**
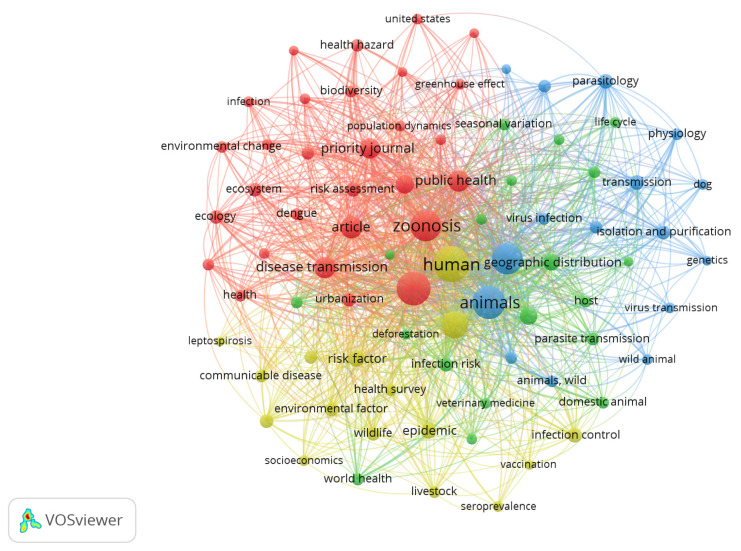
VOSviewer visualization of index keywords.

**Figure 8 ijerph-19-00893-f008:**
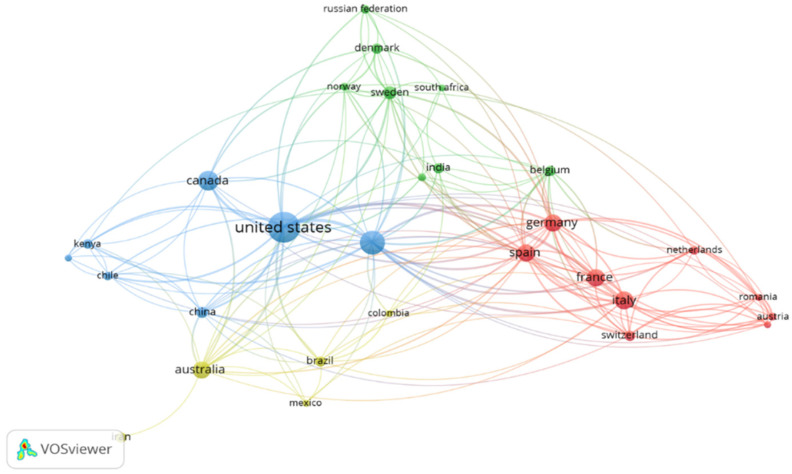
Bibliometric map of the association of countries.

**Figure 9 ijerph-19-00893-f009:**
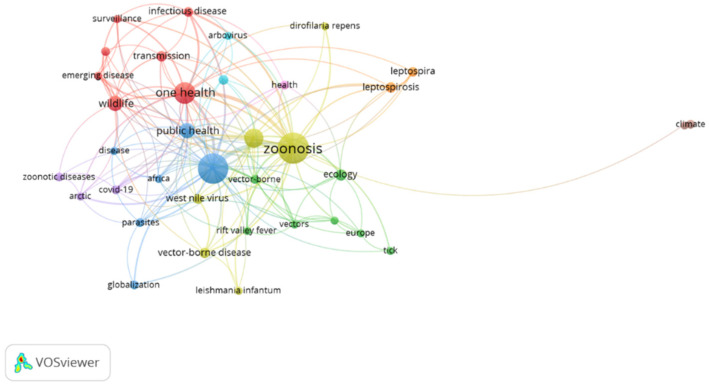
Author keywords.

**Table 1 ijerph-19-00893-t001:** Documents by affiliations.

Rank	Affiliation	Documents
1st	Centers for Disease Control and Prevention	16
2nd	The University of Queensland	10
3rd	London School of Hygiene and Tropical Medicine	10
4th	Harvard University	9
5th	Københavns Universitet	9
6th	Friedrich-Loeffler-Institute	9
7th	Princeton University	8
8th	The Australian National University	8
9th	University of California, Davis	8
10th	University of Saskatchewan	8

**Table 2 ijerph-19-00893-t002:** Documents by country/territory.

Rank	Country/Territory	Documents
1st	United States	130
2nd	United Kingdom	71
3rd	Australia	40
4th	Canada	38
5th	Germany	31
6th	Italy	32
7th	Spain	23
8th	China	20
9th	France	20
10th	India	20

**Table 3 ijerph-19-00893-t003:** Co-authorship by countries.

Rank	Country	Documents	Citations	Total Link Strength
1st	United States	56	5087	53
2nd	United Kingdom	31	3915	43
3rd	Canada	18	1895	32
4th	Australia	18	1314	17
5th	Italy	17	2144	35
6th	Germany	15	826	33
7th	France	11	999	16
8th	Sweden	10	374	20
9th	Spain	9	885	22
10th	The Netherlands	6	976	24

**Table 4 ijerph-19-00893-t004:** Most frequent author keywords.

Rank	Keyword	Frequency	Total Link Strength
1st	Climate change	28	27
2nd	Zoonosis	20	15
3rd	Epidemiology	14	10
4th	One health	12	16
5th	Transmission	7	8
6th	Wildlife	7	8
7th	Bacteria	6	7
8th	Ecology	6	2
9th	Public health	6	4
10th	Climate	5	6

**Table 5 ijerph-19-00893-t005:** Most frequent index keywords per cluster.

	**Red Cluster**			**Green Cluster**		
**No**	**Keyword**	**Occurrences**	**TLS**	**Keyword**	**Occurrences**	**TLS**
1st	climate change	116	1387	Africa	12	164
2nd	Zoonosis	93	1158	animal health	12	173
3rd	Article	50	571	Asia	10	144
4th	public health	47	639	deforestation	10	164
5th	disease transmission	44	610	disease vectors	14	216
6th	priority journal	41	537	domestic animal	14	231
7th	disease carrier	34	488	epidemiology	12	168
8th	risk assessment	22	296	Europe	15	217
9th	Ecosystem	20	285	geographic distribution	29	411
10th	Ecology	18	280	global change	10	151
	**Blue Cluster**			**Yellow Cluster**		
**No**	**Keyword**	**Occurrences**	**TLS**	**Keyword**	**Occurrences**	**TLS**
1st	Animals	107	1315	Human	125	1501
2nd	non-human	95	1249	Review	71	917
3rd	Parasitology	23	332	risk factor	27	384
4th	Transmission	23	327	Epidemic	25	346
5th	Isolation and purification	18	247	infection control	22	293
6th	Microbiology	17	226	Wildlife	20	317
7th	Virology	17	233	communicable diseases, emerging	18	265
8th	disease surveillance	15	213	environmental factor	18	266
9th	animals, wild	14	233	Incidence	17	233
10th	Physiology	14	180	communicable disease	16	202

TLS—Total link strength.

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
