# Peer review of "Climate Change and Zoonoses: A Review of Concepts, Definitions, and Bibliometrics"

_ijerph, 2022, doi:10.3390/ijerph19020893_

Round 1

Reviewer 1 Report

This is an interesting bibliographical-based study examining publications on zoonoses and emerging diseases within a certain time frame and searchable using a specific database. The title and some conclusions overstretch the scope of the study and should be reconsidered to better demonstrate the study results. Some discussion and attention to proper definitions and use of terms (e.g. zoonoses, disease, pathogen, emerging disease, etc) should be addressed earlier in the document. The author's use the term "global crisis" as something that evolved in 2010, but many zoonotic diseases emerged long before that date, which is referring to a spike in specific literature. Undoubtedly there are links between climate change and emerging diseases, but these are not very well presented in the document and a number of other anthropogenic factors associated with emergence of disease (e.g. globilization, trade, and land use change) are not well discussed to differentiate which emerging diseases are attributed directly to climate change (e.g. vector-borne zoonoses) and where there may be less well defined relationships, but likely also ties to anthropogenic causes (e.g. hantaviruses). Some rewording is needed, particularly in the introduction to better explain the background principles and how they relate to the study.  Some specific editorial comments are below: 

Line 17: Consider affects vs effects or re-wording to impact

Line 19: Reword "the West nile and Usutu ones" 

Line 37;610: This sentence seems a bit misleading and it isn't clear what is meant by "which were documented in history," but then evolved in 2010 as that seems to be an oversimplified description of emerging diseases as West Nile emerged at least ten years before that as did brucellosis, tuberculosis, rabies, etc. I suggest providing more clarity here bringing some specific results of this study in terms of numbers of papers that met the search criteria and clearly defining zoonotic disease vs emerging disease and potential links to climate change as many emerging diseases have been linked to anthropogenic changes related to globalization and land-use that aren't necessarily directly linked to climate

Line 45: Do you mean "wildlife" vs untamed life? 

Line 47: This is strangely worded, but I think what is meant here is more the emergence of zoonotic diseases in humans, not just occurrence? Domestic animals have also been a source of many emerging zoonotic diseases without wildlife. 

Line 49-50: Are you referring to antimicrobial resistance here? 

Line 55: This should be zoonotic diseases as human pathogens that are shared with animals not caused by animals as the diseases are caused by pathogens (bacteria, viruses, etc). 

Lines 64-69: Diseases and pathogens are used interchangeably here, but they are differences in interpretation and definition 

Line 72-73: Re-word "maximum impact"

Line 78;93-94;116; 148-149; 180-184; 339-340: All diseases listed here should be lower case (e.g. brucellosis)

Line 88-89: This line seems to contradict other themes in the paper regarding emerging diseases and anticipated changes with climate change as there seems to be an increase in emerging diseases. Please re-word. 

Line 90-91: Update reference and provide more information regarding candidate hosts and understanding of covid origins

Line 92;544: Is there a reference for use of the term "Old zoonoses"; I think what is meant here is that the origin of the pathogen was likely an animal, but it is now maintained in humans, but I am not sure that the term "old zoonoses" is the most appropriate and would suggest a different term

Line 228-229: by "burial sites" are the authors referring to human burial sites? It seems there are concerns about animal mortalities and environmental pathogens (e.g. anthrax bacteria) emerging from permafrost melting sites. Consider re-wording to be more inclusive

Line 530: Scientific names of pathogens should be italicized

Author Response

Thank you for your suggestions. Please find the response in attachment.

Reviewer 2 Report

Reviewed article focused on a bibliometric analysis of effect climate changes and zoonoses for about 20 years is clearly and concisely processed.

This paper will contribute to the dissemination of knowledge about climate changes and zoonoses for threat to human and planetary health.

Author Response

(The authors gave the same response as above.)

Reviewer 3 Report

The review entitled "Climate Change and Zoonoses: a threat to human and planetary health" by Filho et al., provides an interesting analysis of the literature focuses on the research topic "climate change and zoonoses" in the last two decade. The manuscript is very clear and well presented. The introduction describes in great detail the concept of zoonosis and the evolution of the term according to a "one health" approach, in this context the climate change represents a big issue also for the public health and it is closely linked to the spread of 
vectors and consequent trasmissible diseases in free geographic areas. Materials and methods used and the relative crossing of data allows to have a complete overview on the literary production on the subject. As well as the data presented which allow the reader to have a clear picture on the bibliography relating to the topic. The findings give a good idea of the evolution of studies on the subject in relation to climate change, especially in the last decade, with a trend likely to increase in the next years. 
I read this article with interest, I have no considerations to make and I support its publication on IJERPH.

However, in my opinion the title does not convey the idea on the content of the manuscript, it should be clarified that it is a bibliometric analysis on the subject. For this reason I suggest to modified the title.

Author Response

(The authors gave the same response as above.)

Reviewer 4 Report

Climate change can have complex effects that influence human and animal health. This paper reports on an analysis of the literature focused on a bibliometric analysis of the Scopus database and VOSviewer software for creating visualization maps, which identifies the zoonotic health risks for humans and animals caused by climate change. This visualization maps gives the article a novel appearance.

Author Response

(The authors gave the same response as above.)
